# Contact Effects on Thermoelectric Properties of Textured Graphene Nanoribbons

**DOI:** 10.3390/nano12193357

**Published:** 2022-09-27

**Authors:** David M. T. Kuo, Yia-Chung Chang

**Affiliations:** 1Department of Electrical Engineering and Department of Physics, National Central University, Chungli 320, Taiwan; 2Research Center for Applied Sciences, Academic Sinica, Taipei 11529, Taiwan; 3Department of Physics, National Cheng Kung University, Tainan 701, Taiwan

**Keywords:** textured graphene nanoribbons, topological nature, edge states, junction contact, thermoelectric properties, square-form transmission curve

## Abstract

The transport and thermoelectric properties of finite textured graphene nanoribbons (t-GNRs) connected to electrodes with various coupling strengths are theoretically studied in the framework of the tight-binding model and Green’s function approach. Due to quantum constriction induced by the indented edges, such t-GNRs behave as serially coupled graphene quantum dots (SGQDs). These types of SGQDs can be formed by tailoring zigzag GNRs (ZGNRs) or armchair GNRs (AGNRs). Their bandwidths and gaps can be engineered by varying the size of the quantum dot and the neck width at indented edges. Effects of defects and junction contact on the electrical conductance, Seebeck coefficient, and electron thermal conductance of t-GNRs are calculated. When a defect occurs in the interior site of textured ZGNRs (t-ZGNRs), the maximum power factor within the central gap or near the band edges is found to be insensitive to the defect scattering. Furthermore, we found that SGQDs formed by t-ZGNRs have significantly better electrical power outputs than those of textured ANGRs due to the improved functional shape of the transmission coefficient in t-ZGNRs. With a proper design of contact, the maximum power factor (figure of merit) of t-ZGNRs could reach 90% (95%) of the theoretical limit.

## 1. Introduction

Due to global warming, the Kyoto protocol, aiming to reduce CO${}_{2}$ emissions, was proposed in 1997. Since then, renewable energies including solar, wind, rain, tides, and geothermal heat have become topics of tremendous scientific interest [1]. Thermoelectric devices can be used as power generators and refrigerators. The electrical power based on thermoelectric effects is one of the most important types of green energy [1,2,3,4,5,6]. Because of the inherent physics of the thermoelectric effect, thermoelectric devices can fully avoid CO${}_{2}$, hydrofluorocarbon, and perfluorocarbon emissions. In addition, thermoelectric devices can sustain long operation times and avoid mechanical noise.

Recently, energy harvesting applications in nanoscale systems have attracted considerable attention [7,8,9,10,11,12]. The question of how to obtain the maximum thermoelectric efficiency of heat engines with optimized electrical power output has been a key issue [9,10,11,12]. To improve the performance of a heat engine, it is preferable to have the electron transport in the ballistic regime and phonon transport in the diffusive scattering regime. Therefore, a high-performance thermoelectric device needs to provide a channel length shorter than the electron mean free path ($\lambda_{e}$) but much longer than the phonon mean free path ($\lambda_{ph}$) in order to reduce the ratio of phonon thermal conductance ($\kappa_{ph}$) to electron thermal conductance ($\kappa_{e}$) [13]. It was pointed out in Ref. [9] that a Carnot heat engine favors the electron transport in an energy range where the transmission coefficient has a steep change with respect to energy, e.g., with a square form (SF). Up to now, it remains unclear how to realize a SF transmission coefficient in realistic thermoelectric devices with a short channel length between thermal contacts [11,14].

Considerable scientific efforts have paved the way to solving such an intriguing problem. Hicks and Dresselhaus theoretically demonstrated that the thermoelectric performance can be significantly enhanced in one-dimensional (1D) nanowires due to the reduced phonon thermal conductance and the enhanced Seebeck coefficient (*S*) [2,3]. According to theoretical modeling, $\kappa_{ph}$ of 1D silicon quantum dot (QD) superlattices can be reduced by one order of magnitude in comparison with 1D nanowires [7,8]. Due to limitations in technology, the dot-size fluctuation in the 1D silicon quantum dot arrays still remains a serious issue [15,16].

The discovery of graphene in 2004 opened the door to realizing 1D nanowires with a small cross-section to degrade $\kappa_{ph}$ [17], since one can fabricate graphene nanoribbons (GNRs) with atomic precision via the bottom-up approach [18]. This approach has been successfully applied to build more complex systems, such as armchair GNRs with periodically corrugated edges, called textured AGNRs (t-AGNRs) here [19,20,21,22,23,24,25]. The scanning tunneling microscopy (STM) spectra of serially coupled graphene quantum dots (SGQDs) synthesized by t-AGNRs were reported experimentally [25]. Novel graphene-based electronic devices have also focused on AGNRs and t-AGNRs [26]. Recently, the existence of edge states with topological protection in textured zigzag GNRs (t-ZGNRs) has been demonstrated theoretically [27].

Several experimental studies of the thermoelectric properties of graphene-related materials have been reported in recent years [28,29,30,31,32,33,34]. Nevertheless, there is a paucity of studies to consider the contact effect on the thermoelectric properties of t-AGNRs and t-ZGNRs [35]. In this work, our goal is to optimize the transmission coefficient of t-GNRs with length shorter than $\lambda_{e}$, but larger than $\lambda_{ph}$, under different coupling strengths with the electrical contact. We found that minibands can be formed in these t-GNRs, while their energy gaps can be tuned by varying the size of GQDs and the inter-dot coupling strength. Furthermore, the calculated power factor is very robust against scattering from defects occurring inside the interiors of these t-ZGNRs. The transmission coefficients through the minibands of t-ZGNRs under optimized tunneling rates provide desired characteristics showing a steep change with respect to the energy near the central gap that mimics the theoretical limit obtained by using a SF transmission coefficient. Due to this feature, our calculations reveal that the thermoelectric performance of t-ZGNRs can be significantly better than that of t-AGNRs. The maximum power factor (figure of merit) of t-ZGNRs could reach 90% (95%) of that obtained with the SF transmission coefficient. Therefore, the t-ZGNR is a promising candidate for applications in nanoscale energy harvesting.

## 2. Calculation Methods

To model the thermoelectric properties of t-GNRs connected to the electrodes, it is a good approximation to employ a tight-binding model with one $p_{z}$ orbital per atomic site to describe the electronic states [36,37,38,39,40]. The Hamiltonian of the nano-junction system depicted in Figure 1, including four different GNR structures, can be written as $H=H_{0}+H_{GNR}$ [41], where
(1)$$\begin{array}{ccc} H_{0} & = & \sum_{k}\epsilon_{k}a_{k}^{†}a_{k}+\sum_{k}\epsilon_{k}b_{k}^{†}b_{k}\\ & + & \sum_{\ell }\sum_{k}V_{k,\ell ,j}^{L}d_{\ell ,j}^{†}a_{k}+\sum_{\ell }\sum_{k}V_{k,\ell ,j}^{R}d_{\ell ,j}^{†}b_{k}+h.c. \end{array}$$

The first two terms of Equation (Equation 1) describe the free electrons in the left (*L*) and right (*R*) electrodes. $a_{k}^{†}$ ($b_{k}^{†}$) creates an electron with wave number *k* and energy $\epsilon_{k}$ in the left (right) electrode. $V_{k,\ell ,j=1}^{L}$ ($V_{k,\ell ,j=N_{z}\left(N_{a}\right)}^{R}$) describes the coupling between the left (right) lead with its adjacent atom in the *ℓ*-th row.
(2)$$\begin{array}{ccc} H_{GNR} & = & \sum_{\ell ,j}E_{\ell ,j}d_{\ell ,j}^{†}d_{\ell ,j}-\sum_{\ell ,j}\sum_{\ell^{\prime },j^{\prime }}t_{\left(\ell ,j\right),\left(\ell^{\prime },j^{\prime }\right)}d_{\ell ,j}^{†}d_{\ell^{\prime },j^{\prime }}+h.c, \end{array}$$
where $E_{\ell ,j}$ is the on-site energy for the $p_{z}$ orbital in the *ℓ*-th row and *j*-th column. Here, the spin-orbit interaction is neglected. $d_{\ell ,j}^{†}\left(d_{\ell ,j}\right)$ creates (destroys) one electron at the atom site labeled by (*ℓ*,*j*), where *ℓ* and *j*, respectively, are the row and column indices as illustrated in Figure 1. $t_{\left(\ell ,j\right),\left(\ell^{\prime },j^{\prime }\right)}$ describes the electron hopping energy from site (*ℓ*,*j*) to site ($\ell^{\prime }$,$j^{\prime }$). The tight-binding parameter used for GNRs is $E_{\ell ,j}=0$ for the on-site energy and $t_{\left(\ell ,j\right),\left(\ell^{\prime },j^{\prime }\right)}=t_{pp\pi }=2.7$ eV for the nearest-neighbor hopping strength.

To study the transport properties of a GNR junction connected to electrodes, it is convenient to use the Keldysh Green function technique [41]. Electron and heat currents leaving the electrodes can be expressed as
(3)$$\begin{array}{ccc} J & = & \frac{2e}{h}\int d\varepsilon \;T_{LR}\left(\varepsilon \right)\left[f_{L}\left(\varepsilon \right)-f_{R}\left(\varepsilon \right)\right], \end{array}$$
and
(4)$$Q_{e,L(R)}=\frac{\pm 2}{h}\int d\varepsilon \;T_{LR}\left(\varepsilon \right)\left(\varepsilon -\mu_{L(R)}\right)\left[f_{L}\left(\varepsilon \right)-f_{R}\left(\varepsilon \right)\right]$$
where $f_{\alpha }\left(\varepsilon \right)=1/\left\{\exp\left[\left(\varepsilon -\mu_{\alpha }\right)/k_{B}T_{\alpha }\right]+1\right\}$ denotes the Fermi distribution function for the α electrode, where $\mu_{\alpha }$ and $T_{\alpha }$ are the chemical potential and the temperature of the α electrode. *e*, *h*, and $k_{B}$ denote the electron charge, the Planck constant, and the Boltzmann constant, respectively. $T_{LR}\left(\varepsilon \right)$ denotes the transmission coefficient of a GNR connected to electrodes, which can be solved by the formula $T_{LR}\left(\varepsilon \right)=4Tr\left[\Gamma_{L}\left(\varepsilon \right)G^{r}\left(\varepsilon \right)\Gamma_{R}\left(\varepsilon \right)G^{a}\left(\varepsilon \right)\right]$ [42,43], where $\Gamma_{L}\left(\varepsilon \right)$ and $\Gamma_{R}\left(\varepsilon \right)$ denote the tunneling rate (in energy units) at the left and right leads, and $G^{r}\left(\varepsilon \right)$ and $G^{a}\left(\varepsilon \right)$ are the retarded and advanced Green’s functions of the GNR, respectively. The tunneling rates are described by the imaginary part of the self-energy correction on the interface atoms of the GNR due to the coupling with nearby atoms in the adjacent electrodes, i.e., $\Gamma_{L(R)}\left(\varepsilon \right)=-Im(\Sigma_{L(R)}^{r}\left(\varepsilon \right)$). Such coupling depends on the contact quality with the electrodes, which is characterized by the interaction strength $V_{k,\ell ,j=1}^{L}\left(V_{k,\ell ,j=N_{z}\left(N_{a}\right)}^{R}\right)$ with the left (right) lead. Here, we have adopted energy-independent tunneling rates $\Gamma_{L(R)}\left(\varepsilon \right)=\Gamma_{L(R)}$, which is reasonable in the wide-band limit for the electrodes [42]. Note that $\Gamma_{\alpha }$ and Green’s functions are matrices in the basis of tight-binding orbitals. $\Gamma_{\alpha }$ for the boundary atoms have diagonal entries given by the same constant $\Gamma_{t}$. Because of the line contact, as illustrated in Figure 1, the contact resistance can be much smaller than that of surface contact [44]. Meanwhile, the variation of tunneling rates could reveal different contact properties, such as the Schottky or ohmic contact [45].

In the linear response regime, the electrical conductance ($G_{e}$), Seebeck coefficient (*S*), and electron thermal conductance ($\kappa_{e}$) are given by $G_{e}=e^{2}L_{0}$, $S=-L_{1}/\left(eTL_{0}\right)$, and $\kappa_{e}=\frac{1}{T}\left(L_{2}-L_{1}^{2}/L_{0}\right)$, with $L_{n}$ (=0,1,2) defined as
(5)$$L_{n}=\frac{2}{h}\int d\varepsilon \;T_{LR}\left(\varepsilon \right){\left(\varepsilon -\mu \right)}^{n}\frac{\partial f(\varepsilon )}{\partial \mu }.$$

Here, $f\left(\varepsilon \right)=1/\left(exp^{\left(\varepsilon -\mu \right)/k_{B}T}+1\right)$ is the Fermi distribution function of electrodes at equilibrium temperature *T* and chemical potential μ. As seen in Equation (Equation 5), the transmission coefficient $T_{LR}\left(\varepsilon \right)$ plays a significant role for electron transport between the left (*L*) and right (*R*) electrodes. At zero temperature, the electrical conductance is given by $G_{e}\left(\mu \right)=\frac{2e^{2}}{h}T_{LR}\left(\mu \right)$. In the current study, only even numbers of $N_{a}$ are considered, to avoid unwanted dangling-bond states. We have developed an efficient computation method that allows us to calculate the non-equilibrium Green’s functions of large-sized quantum structures. For the current study, textured ZGNRs and AGNRs with lengths up to 14 nm are considered.

## 3. Results and Discussion

### 3.1. Graphene Nanoribbons

Topological states offer promising applications in electronics and optoelectronics, owing to their robustness in transport characteristics against defect scattering. Many studies have confirmed that 2D and 1D topological states (TSs) exist in certain material structures [46,47,48,49,50,51]. For instance, there exists a 2D topologically protected interface state in HgTe/CdTe superlattices [46,47]. One-dimensional (1D) TSs were theoretically predicted to exist in square selenene and tellurene [51]. Recently, zero-dimensional (0D) TSs of finite-size GNRs have been extensively studied [19,20,21,22,23,24,25] because the 0D TS offers more flexibility in the design of complicated electronic circuits [22]. Before illustrating electron-coherent transport through SGQDs formed by t-ZGNRs, we first examine the characteristics of 0D TSs of finite-size GNRs by calculating their transmission coefficient, $T_{LR}\left(\varepsilon \right)$.

Figure 2 shows the calculated electron conductance, $G_{e}$, at $k_{B}T=0$ as a function of μ for various $N_{a}$ with $N_{z}=7$. We note that there are two zigzag edge states localized at the top and bottom sides of the GQDs depicted in Figure 1a. For $N_{a}=40$, their wave functions are well separated along the armchair direction [37]. The electrical conductance spectrum shows that $G_{e}=2G_{0}$ at μ=0 and $G_{e}=G_{0}$ for other electronic states, where $G_{0}=\frac{2e^{2}}{h}$ is the quantum conductance. Such zero energy modes were observed experimentally by STM [23]. When two electronic states are closely spaced (with energy separation less than the broadening), $G_{e}$ can become larger than $G_{0}$. As $N_{a}$ decreases, $G_{e}$ for the zero-energy mode is split into two peaks corresponding to the bonding and antibonding states of coupled zigzag edge states, as seen in Figure 2c. When $N_{a}=12$, these two peaks are well separated. Here, $\epsilon_{HO}$ and $\epsilon_{LU}$ denote the energy levels of the highest occupied molecular orbital (HOMO) and the lowest unoccupied molecular orbital (LUMO), respectively. Due to the short decay lengths along the armchair edge directions for zigzag edge states, $T_{LR}\left(\varepsilon \right)$ of $\Sigma_{0}$ also depends on whether the zigzag or armchair edges are coupled to the electrodes. In Ref. [52], it is proposed that a single QD can be used to realize a Carnot heat engine. However, the channel length for $N_{z}=7$ ($L_{z}=0.738$ nm) is too small to avoid the serious degradation of the figure of merit due to the phonon heat conductivity, $\kappa_{ph}$. On the other hand, for the large $N_{z}$ case ($N_{z}\gg N_{a}$), a finite GNR shows metallic behavior, leading to unfavorable thermoelectric properties. To maintain a sizable gap (10 $k_{B}T$) around the charge neutrality point (CNP) while keeping $\kappa_{ph}$ small enough to preserve a decent figure of merit ($ZT=\frac{S^{2}G_{e}T}{\kappa_{e}+\kappa_{ph}}$), t-ZGNR becomes a suitable candidate, as we shall discuss below.

### 3.2. SGQDs Formed by Textured ZGNRs

For energy harvesting applications at room temperature, we need to design a $T_{LR}\left(\varepsilon \right)$ spectrum with a square shape near the central gap to obtain optimized ZT and electrical power output [9]. Let us consider an SGQD formed by t-ZGNR. A t-ZGNR can be realized by tailoring a GNR with periodic indentation on the zigzag edges, such as the structure shown in Figure 1b. This t-ZGNR consists of GQDs with size characterized by $N_{a}=12$ and $N_{z}=7$. A single GQD of this size has resonance energies $\varepsilon_{LU(HO)}=\pm 0.247$ eV near the CNP, as shown in Figure 2c. For two coupled GQDs (2GQDs), there is one satellite peak on the left (right) side of the $\varepsilon_{HO(LU)}$ peak (see Figure 3b). For six coupled GQDs (6GQDs), we have five additional peaks with $\varepsilon_{e(h),1}=\pm 0.3105$ eV, $\varepsilon_{e(h),2}=\pm 0.468$ eV, $\varepsilon_{e(h),3}=\pm 0.6705$ eV, $\varepsilon_{e(h),4}=\pm 0.882$ eV, and $\varepsilon_{e(h),5}=\pm 1.0665$ eV on the right (left) side of $\varepsilon_{LU(HO)}=\pm 0.247$ eV (see Figure 3f). We note that $G_{e}\left(\mu \right)$ is fully suppressed for μ between $\varepsilon_{HO}$ and $\varepsilon_{LU}$. Therefore, SGQDs formed by t-ZGNRs function as filters to block electrons with energies near the CNP, which is different from that of serially coupled antidots realized by ZGNRs with nanopores [53]. The separation between peaks is inhomogeneous. In addition, the $\varepsilon_{HO}$ and $\varepsilon_{LU}$ peaks become sharper with increasing GQD number. These features are attributed to a special parity symmetry in the supercell of ZGNRs. Due to the coupling of transverse and longitudinal wave numbers in ZGNRs, the density distributions of the electronic state near CNP are inhomogeneous [54]. AGNRs do not have such a parity in the supercell. The feature provides a significant effect on the transmission coefficient of SGQDs formed by t-ZGNRs. To further illustrate the characteristics of the edge states, we show the charge densities of states with $\varepsilon_{LU}$, $\varepsilon_{e,1}$, and $\varepsilon_{e,2}$ for the case of 6GQDs in Figure A1 of Appendix A.

Previous theoretical studies have demonstrated that edge defects can significantly reduce the electron quantum conductance of ZGNRs [55,56]. Here, we investigate how defects (either at the edge or in the interior region) influence the thermoelectric characteristics of t-ZGNRs by introducing energy shifts $\Delta_{\ell ,j}$ on designated defect sites. $\Delta_{\ell ,j}$ could be positive or negative, which depends on the type of defect [56]. To model a vacancy within a tight binding model, one typically takes $\Delta_{\ell ,j}\to \infty$. The larger the orbital energy shift, the stronger the effect on the electrical conductance [57]. Here, we consider the case of a positive and large Δ to investigate the effects of defects on the electron transport of t-ZGNRs. We calculate $G_{e}$, $\kappa_{e}$, *S*, and power factor $PF=S^{2}G_{e}$ for different defect locations for the case of 15 GQDs ($L_{z}=14.5$ nm or $N_{z}=119$) with $\Gamma_{t}=0.54$ eV and show the results in Figure 4.

As seen in Figure 4a,b, the size of the central gap and the shape of $G_{e}$ and $\kappa_{e}$ spectra (green lines) are changed only slightly when the defects are located in the interior region. However, the $G_{e}$ and $\kappa_{e}$ values near the central gap become seriously suppressed (as shown by red lines) when a single defect is located at a zigzag edge site labeled by (1,4), where the charge density of the $\varepsilon_{LU(HO)}$ electronic state is peaked. As seen in Figure 4c, the antisymmetric pattern of the Seebeck coefficient S(μ) (with respect to the sign change of μ) due to the electron–hole symmetry is distorted in the presence of defects. The peak value of the power factor $PF=S^{2}G_{e}$ near the central gap is seriously reduced when defects occur on the edge, but much less affected by defects in the interior region, as illustrated in Figure 4d. Here, and henceforth, $\kappa_{e}$ is in units of $\kappa_{0}=0.62$ nW/K, *S* is expressed in units of $k_{B}/e=86.25$μV/K, and the power factor (PF) in units of $2k_{B}^{2}/h=0.575$ pW/K${}^{2}$. It is remarkable to see that PF is very robust against defect scattering when defects are away from zigzag edges. Such location-dependent effects can be depicted by the charge density distribution in Figure A1. To reduce the defect effect on electron transport, one needs to avoid creating defects randomly located on the zigzag edges.

We note that SGQDs are formed by periodically removing some carbon atoms on the zigzag edges of ZGNRs. To understand the relationship between the quantum states near the central gap in the SGQD formed by t-ZGNRs and the quantum states near the zero energy modes of ZGNRs, we show a comparison of the subband structures of the infinitely long t-ZGNR shown in Figure 1b and unaltered ZGNR shown in Figure 1a (with an enlarged supercell of length L=4a to match the unit cell of t-ZGNR) in Figure A2a of the Appendix A. Note that the first Brillouin zone (BZ) has one quarter the size of the BZ of ZGNR. Thus, the subband structures shown in Figure A2a include the zone-folding effect that maps the zone boundary ($k_{z}=\pi /a$) of the unfolded BZ of ZGNR to $k_{z}=0$ of the folded BZ here. As expected, the unaltered ZGNR has edge states with zero energy for $k_{z}<0.06$ π/a=0.24 π/L. The two edge states are strongly coupled in the t-ZGNR, with energy splitting of 2.5 eV. On the other hand, the edge states at the mini-zone boundary ($k_{z}=$
π/L) with energies corresponding to the HOMO (LUMO) level are nearly unaffected by the removal of a carbon atom in the neck region of t-ZGNR, since the wave functions of these edge states have a node at that position. Theoretical calculations of the Zak phases of various t-ZGNRs indicate that the topological nature of the edge states of t-ZGNRs is preserved even though their energy levels are shifted away from the CNP [27]. Thus, the transport through these states should be rather insensitive to the presence of defects inside the t-ZGNR.

Junction tunneling rates can be affected by the quality of contact between the electrode and semiconductor [44], which is a critical issue for device applications of two-dimensional materials [58]. To clarify the contact effect, we show in Figure 5 the calculated $G_{e}$, $\kappa_{e}$, *S*, and ZT at room temperature for various tunneling rates ($\Gamma_{t}$) for a t-ZGNR with 15 GQDs, each having the dimension $N_{a}=8$ and $N_{z}=7$, as shown in Figure 1b. When the tunneling rate increases, the transmission coefficient $T_{LR}\left(\epsilon \right)$ through minibands close to HOMO (LUMO) is enhanced. For the case of 15 GQDs, there are 15 peaks in each miniband (see Figure A3). Such an enhancement leads to an increase in both $G_{e}$ and $\kappa_{e}$, as seen in Figure 5a,b. However, *S* is essentially independent of the variation in $\Gamma_{t}$, as indicated by the collapsing of all four curves in Figure 5c. As a result, the power factor $PF=S^{2}G_{e}$ also increases with $\Gamma_{t}$. It is noted that the maximum S reaches 1.51 mV/K (see Figure 5c), which is much larger than that observed experimentally in gapless graphene [28,29]. In Figure 5d, the maximum ZT occurs at μ=±0.423 eV, where $\kappa_{ph}\gg \kappa_{e}$. Therefore, the enhancement in ZT with respect to the increase in $\Gamma_{t}$ mainly arises from the enhancement in PF. Here, we have included the effect of phonon thermal conductance, $\kappa_{ph}$, and assumed $\kappa_{ph}=F_{s}\kappa_{ph}^{0}$ in the calculation of ZT in Figure 5d, where $\kappa_{ph}^{0}=\frac{\pi^{2}k_{B}^{2}T}{3h}$ is the phonon thermal conductance of an ideal ZGNR. We adopt the room-temperature value of $\kappa_{ph}^{0}=0.285$ nW/K for the ZGNR with width $N_{a}\leq 8$ obtained by a first-principles calculation as given in Ref. [59]. $F_{s}=0.1$ denotes a reduction factor resulting from quantum constriction in t-ZGNRs. It has been theoretically demonstrated that the magnitude of $\kappa_{ph}$ can be reduced by one order magnitude for ZGNRs with “edge vacancies” [60], which is similar to the mechanism introduced in silicon nanowires with surface roughness [61]. The measured $\lambda_{ph}$ can be reduced from 300–600 nm in a single layer graphene to 10 nm in graphene nanostructures (see [30] and references therein). Recently, a very short $\lambda_{ph}$ has been reported experimentally, which offers promise for enhancing the figure of merit (ZT) of graphene heterostructures [31,32,33,34].

The calculated results shown in Figure 5 imply that ZT>3 could be realized by using SGQDs formed by t-ZGNRs for tunneling rates corresponding to $\Gamma_{t}$ between $t_{pp\pi }/10(0.27$ eV) and $t_{pp\pi }/5(0.54$ eV). In Figure A3 of Appendix A, we show the peak value of $ZT(ZT_{max})$ as a function of tunneling rate $\Gamma_{t}$, and we found that the $ZT_{max}$ can be larger than 3 for $\Gamma_{t}$ between 0.18 eV and 1.45 eV. We note that if a semi-infinite ZGNR contact is used, it is reasonable to assume that the coupling strength between the semi-infinite ZGNR and t-ZGNR can be comparable to the hopping strength $t_{pp\pi }=2.7$ eV. Here, we show that the maximum ZT can reach 3.7 with $\Gamma_{t}=0.54$ eV = $t_{pp\pi }/5$, which could be achievable with good contact.

Because *S* is a robust physical quantity with respect to the variation in tunneling rate and channel length (which only depend on the magnitude of the central gap and temperature in our case), the optimization of $G_{e}$ becomes a critical issue in SGQDs when $\kappa_{ph}\gg \kappa_{e}$. To provide a better understanding of the effect of the tunneling rate on the thermoelectric properties, we further investigate the relation between the tunneling rate and the spectral shape of $T_{LR}\left(\varepsilon \right)$. In Figure 6, we show a comparison of the tunneling spectra, $T_{LR}\left(\varepsilon \right)$, of a t-ZGNR with 15 GQDs calculated for two different tunneling strengths, $\Gamma_{t}=0.54$ eV and $\Gamma_{t}=t_{pp\pi }=2.7$ eV. In Figure 6a, the area below the $T_{LR}\left(\varepsilon \right)$ curve shows a right-triangle shape that has a steep change with respect to ε on the side toward the central gap. In Figure 6b (with $\Gamma_{t}=t_{pp\pi }=2.7$ eV), the electron tunneling probability through the electronic states near $\varepsilon_{LU(HO)}$ is highly suppressed, leading to an arch-like area under the $T_{LR}\left(\varepsilon \right)$ curve. The corresponding maximum ZT for this case is 2.3, which is much smaller than that of Figure 5d. The results of Figure 6a,b indicate that the shape of the area under the $T_{LR}\left(\varepsilon \right)$ curve depends on the tunneling rate (or the quality of contact). An arch-like area under $T_{LR}\left(\varepsilon \right)$ will reduce the electrical conductance $G_{e}$ near the central gap.

As seen in Figure 6c, the optimized PF found near $\Gamma_{t}=0.54$ eV is very close to that obtained by using an ideal SF transmission spectrum (indicated by the red line), which exhibits the quantum limit of power factor for 1D systems with $PF_{QB}=1.2659\left(\frac{2k_{B}^{2}}{h}\right)$ [9,11]. We obtain the optimized $PF_{max}=0.9$$PF_{QB}$, which is the same as that of the quantum interference heat engine [62]. This analysis suggests that using SGQDs with suitable tunneling rates could achieve performance close to an optimum heat engine with maximum electrical power output and high thermoelectric efficiency. With the same $\kappa_{ph}$, the $ZT_{max}$ of t-ZGNRs can reach 95% of that obtained by an SF transmission coefficient (with $ZT_{max}=3.951$). Thus far, we have only considered periodically indented structures on both the top and bottom zigzag edges (see Figure 1b). We note that the same optimized result of $PF_{max}=0.9$$PF_{QB}$ can also be achieved in periodically indented structures corresponding to Figure 6 but with a textured pattern only on one zigzag side, provided that the tunneling rate can be increased to $\Gamma_{t}=0.72$ eV. The results are shown in Figure A3 of Appendix A.

### 3.3. Armchair Graphene Nanoribbons

As shown in Figure 5 and Figure 6, SGQDs formed by metallic ZGNRs can become semiconductors. Next, we investigate whether their thermoelectric performance is better than that of AGNRs or t-AGNRs. We note that many designs have focused on the optimization of the thermoelectric performance of AGNRs [63,64,65,66]. Here, we first consider AGNRs with their zigzag ends coupled to electrodes, as depicted in Figure 1c. Figure 7 shows the calculated transmission coefficient $T_{LR}\left(\epsilon \right)$ for different defect locations with $N_{z}=7$, $N_{a}=100$ and Δ=5.4 eV. We choose $\Gamma_{t}=2.7$ eV, which gives the optimized shape of the transmission coefficient in a defect-free situation in Figure 7a. For finite-size AGNRs, the conduction (valence) subband states are quantized, leading to closely spaced peaks with staircase-like structures, as revealed by the $T_{LR}\left(\varepsilon \right)$ spectrum. The area under the $T_{LR}\left(\varepsilon \right)$ curve for states derived from the first subband has a parabolic shape, which does not meet the criterion for achieving an optimized thermoelectric property. For a single defect at edge location (1,3), all but the transmission coefficient spectrum of the first conduction subband are affected significantly. When the defect is located at site (3,3), the spectra for the first conduction subband and the first valence subband are nearly unaffected. Nevertheless, when the defect occurs at (4,4) the spectra in these two subbands show a remarkable change. In the current studies, we do not observe the Anderson localization effect, which leads to a vanishing $G_{e}$ in the subband regions when a single defect occurs on the edges of ZGNRs and AGNRs [67]. This implies that the GNRs considered are not exact 1D systems.

To illustrate the effects of defects on the thermoelectric properties of finite AGNRs, we show the calculated electrical conductance ($G_{e}$), Seebeck coefficient (*S*), power factor (PF), and figure of merit (ZT) as functions of μ at $k_{B}T=27$ meV in Figure 8. The four curves in each diagram correspond to the effects of defects located at the four sites considered in Figure 7. As seen in Figure 8a, the spectra of $G_{e}$ for defects at locations (1,3) and (4,4) are degraded seriously, either in the valence subband or conduction subband, producing highly asymmetrical behavior in electron transport. Due to the robustness of the Seebeck coefficient in the central gap region, the effect of defects on the power factor is solely determined by $G_{e}$. Although the maximum *S* is larger than that of Figure 5c due to the larger gap in AGNRs with $N_{z}=7$, the PF values are smaller than the corresponding values of the optimized t-ZGNR shown in Figure 6 (with $\Gamma_{t}=0.54$ eV). This is mainly attributed to the different shapes in the area under the transmission coefficient curve. As a consequence, the $G_{e}$ in AGNRs resulting from thermionic tunneling effects is less favorable. Note that we have adopted $\kappa_{ph}=0.0285$ nW/K [59] in Figure 8d, which is the same as the $\kappa_{ph}$ used in Figure 5d.

### 3.4. SGQDs Formed by Textured AGNRs

Finally, we study the thermoelectric effect of SGQD structures based on the t-AGNR structure, as shown in Figure 1d. The electronic structures and density of states (DOS) of t-AGNR superlattices have been studied theoretically by using density functional theory (DFT) [21,40]. Electronic structures of t-AGNRs can also be calculated by using a tight-binding model, and results are in good agreement with the DFT calculation [40]. Figure 9 shows the calculated $T_{LR}\left(\varepsilon \right)$ of SGQD junctions made from t-AGNRs as functions of ε for three different $N_{a}$ values with $N_{z}$ fixed at 5. We adopt the optimized tunneling rate with $\Gamma_{t}=2.7$ eV. Although the AGNR with $N_{z}=5$ is metallic, the textured AGNR can be semiconducting, as illustrated by the sizable central gap (1.098 eV) in Figure 9. In Figure A2b of Appendix A, we show a comparison of the subband structures of the t-AGNR and unaltered AGNR (with an enlarged supercell of length $L=2a^{\prime }$ to match the unit cell of t-AGNR), where $a^{\prime }=\sqrt{3}a$ is the unit cell length of the AGNR. We see that the two subbands with linear dispersion in the metallic $N_{z}=5$ AGNR are split into two subbands with parabolic dispersion near zero wave number once the quantum constriction takes effect in the t-AGNR. It is worth noting that edge states with zero energy can exist at the left and right ends (with zigzag edges) of the truncated t-AGNR in contact with electrodes. These edge states can contribute to $T_{LR}\left(\varepsilon \right)$ for t-AGNRs with a short SGQD structure (see Figure A4 in Appendix A), since their wave functions decay exponentially along the armchair direction. The area under the $T_{LR}\left(\varepsilon \right)$ curve has an arch-like shape that does not change much as we vary the tunneling rate.

Figure 10 shows the calculated $G_{e}$, *S*, PF, and ZT of SGQD junctions made from a t-AGNR with $N_{z}=5$ and $N_{a}=132$ ($N_{QD}=15$) as functions of temperature for different μ values. Solid and dashed lines correspond to μ=0.52 eV and μ=0.36 eV, respectively. The first conduction subband edge occurs at $\mu_{edge}=0.544$ eV. At a given temperature $k_{B}T=27$ meV, the maximum power factor occurs at μ=0.52 eV. For μ=0.36 eV, which is far away from $\mu_{edge}$; the calculated $G_{e}$ in the thermionic-assisted tunneling process (TATP) is extremely small, whereas its Seebeck coefficient is highly enhanced. The temperature-dependent *S* for this case is complicated. Its temperature dependence can be described by three different functions in three temperature ranges, as illustrated in Figure 10b. In region one (T<90 K), we have $S_{1}=-\frac{\pi^{2}k_{B}^{2}T}{3e}\frac{\partial ln(T_{LR}\left(\varepsilon \right))}{\partial \varepsilon }{|}_{\varepsilon =\mu }$. In region three (T>120 K), we have $S_{3}=\frac{\mu -\mu_{edge}}{eT}$ [68]. In region two, the analytic expression of $S_{2}$ is unknown. In TATP, $G_{e}$ can be described by the expression $exp^{\left(\mu -\mu_{edge}\right)/\left(k_{B}T\right)}$. As a consequence, the power factor and figure of merit are small for μ=0.36 eV. When μ=0.52 eV, the PF is enhanced quickly in the range 50 K < *T* < 120 *K*. Within this temperature region, ZT shows a similar behavior of PF. When T>120 K, the temperature-dependent ZT has the same trend of *S* because the heat current is dominated by the linear-*T* phonon thermal conductance $\kappa_{ph}=F_{s}\frac{\pi^{2}k_{B}^{2}T}{3h}$, which cancels out the factor of $TG_{e}$ in the numerator of ZT. It is worth noting that the optimized PF of t-AGNRs at T=325 K is only one half of the power factor of the t-ZGNR shown in Figure 6c.

Overall, we found no appreciable improvement in the power factor of t-AGNRs in comparison to AGNRs. Although electron Coulomb interactions are neglected in this calculation, our conclusion is still valid even in the Coulomb blockade regime. When the thermoelectric behavior of SGQDs is dominated by the TATP, electron–electron correlation functions are usually small [69]. Therefore, we can neglect electron Coulomb interactions when μ is inside the gap between two subbands.

## 4. Conclusions

We have theoretically investigated the transport and thermoelectric properties of SGQDs, which are formed by tailoring ZGNRs and AGNRs. Our calculations are based on Green’s function approach within a tight-binding model. An electron-coherent tunneling process is found to be responsible for the electrical conductance spectra of SGQDs. The subband width and central gap of SGQDs can be modulated by varying the size of GQDs and inter-dot coupling strength. Unlike $G_{e}$ and $\kappa_{e}$, Seebeck coefficients are found to be insensitive to the contact property between the SGQDs and electrodes. As a result, the power factor and thermoelectric figure of merit can be improved by modulating the tunneling rates. The maximum ZT values at room temperature occur when the chemical potential, μ, is close to the HOMO (LUMO) level in a typical situation where phonons dominate the heat transport. As shown in Figure 6 and Figure A3, the maximum power factor (figure of merit) of t-ZGNRs at room temperature can reach 90% (95%) of the ideal situation with a square-form transmission coefficient. We also found that SGQDs based on t-ZGNRs can outperform SGQDs based on t-AGNRs for thermoelectric applications. The significantly improved thermoelectric behavior of the textured ZGNR is attributed to the sharp change in its transmission coefficient near the central gap. We found that defects at interior sites will not ruin the robust behavior of $T_{LR}\left(\varepsilon \right)$ associated with the edge states near the LUMO of t-ZGNRs. This implies that the electron mean free path, $\lambda_{e}$, contributed by the zigzag edge states ($\lambda_{e,edge}$), can be much larger than the contribution from bulk states ($\lambda_{e,bulk}$). When the channel length ($L_{z}$) of t-ZGNRs is much larger than $\lambda_{ph}$ and $\lambda_{e,bulk}$, $\kappa_{ph}$ is seriously suppressed in such a diffusing region. The electronic states in the first miniband (near LUMO) of t-ZGNRs showing nonlinear dispersion could remain coherent due to their unique nature [27] (see Figure A2a). As a consequence, the power factor of Figure 6 remains valid; meanwhile, the corresponding ZT could be further enhanced [13]. At room temperature with T=324 K ($k_{B}T=27$ meV), the electrical power output can reach 0.212 nW/K for each SGQD implemented by using t-ZGNR. For an SGQD array with a density of $5\times 10^{6}$ cm${}^{-1}$ and ZT larger than 3, the estimated power output is around mW/K, which can be applicable for low-power wearable electronic devices [70].

## Figures and Tables

**Figure 1 nanomaterials-12-03357-f001:**
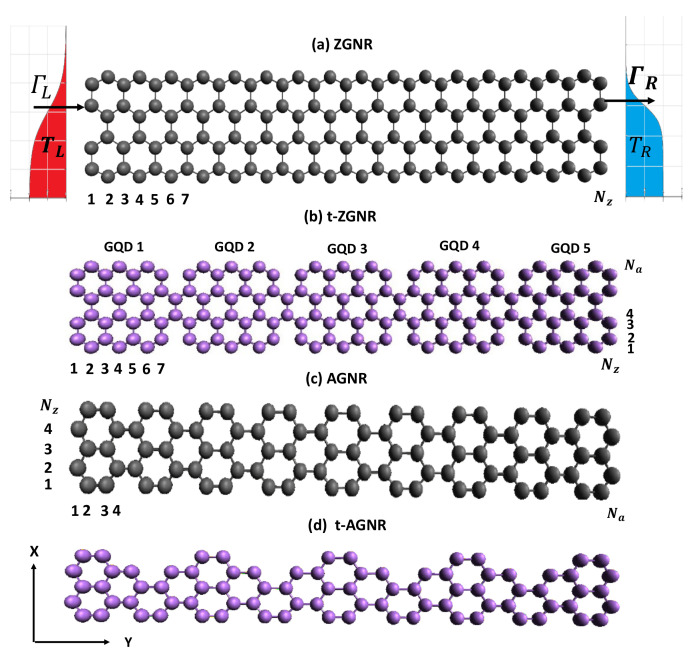
Schematic diagrams of (**a**) ZGNR, (**b**) textured-ZGNR (t-ZGNR), (**c**) AGNR, and (**d**) t-AGNR. In (**a**), we also show the electrodes with $\Gamma_{L}$ ($\Gamma_{R}$) denoting the tunneling rate of the electrons between the left (right) electrode and the leftmost (rightmost) atoms of the ZGNR. $T_{L}$ and $T_{R}$ denote the temperature of the left (*L*) and the right (*R*) electrodes, respectively. Note that t-ZGNRs can be formed by periodically removing some carbon atoms on the zigzag edges of ZGNRs. In (**b**), the t-ZGNR consists of GQDs with size characterized by $\left(N_{a},N_{z}\right)=\left(8,7\right)$. In (**c**), the AGNR has a length $N_{a}=36$ and width $N_{z}=5$. In (**d**), the width and length of the unit cell for the t-AGNR superlattice are characterized by $N_{z}=5$ and $N_{a}=8$, respectively.

**Figure 2 nanomaterials-12-03357-f002:**
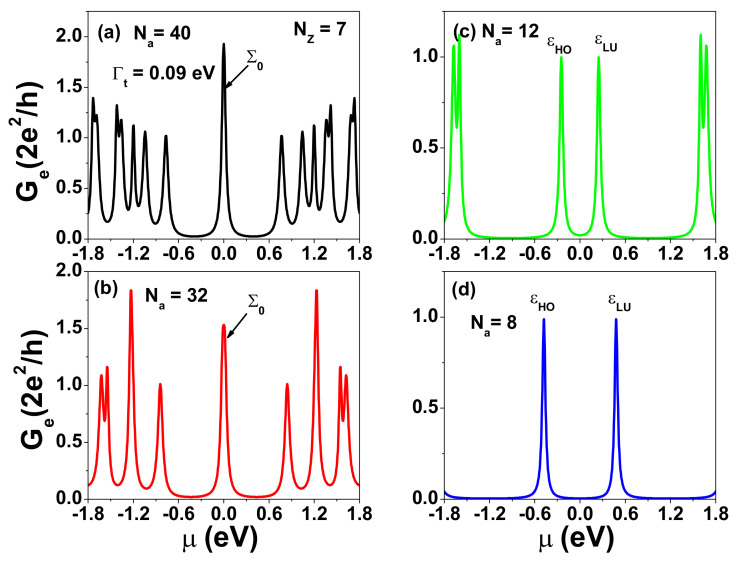
Electrical conductance, $G_{e}$, of a finite GNR with armchair edges coupled to the electrodes at $k_{B}T=0$ as a function of the chemical potential, μ, for various widths with (**a**) $N_{a}=40$, (**b**) $N_{a}=32$, (**c**) $N_{a}=12$, and (**d**) $N_{a}=8$. $N_{z}$ is fixed at 7. Here, we have adopted an electron tunneling rate, $\Gamma_{t}=90$ meV.

**Figure 3 nanomaterials-12-03357-f003:**
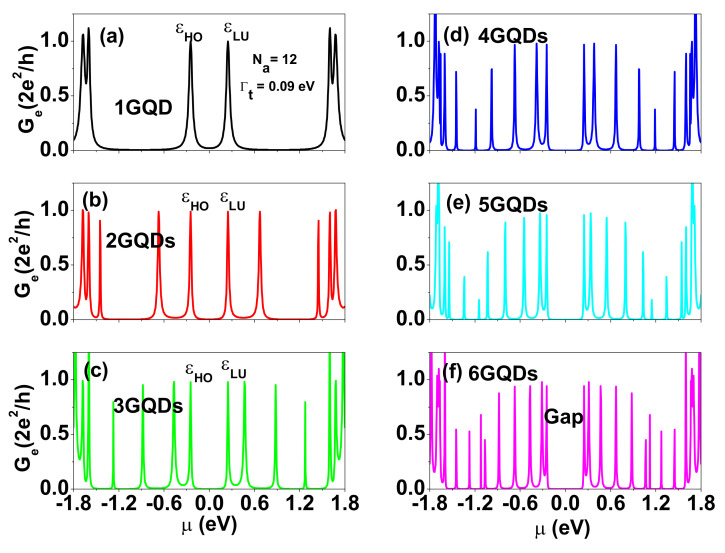
Electrical conductance, $G_{e}$, as a function of μ for SGQDs with GQD numbers, n=1,2,3,4,5, and 6 in subfigures (**a**) 1GQD, (**b**) 2GQDs, (**c**) 3GQDs, (**d**) 4GQDs, (**e**) 5GQDs, and (**f**) 6GQDs, respectively. The size of each GQD in these SGQDs is characterized by $N_{a}=12$ and $N_{z}=7$. Other physical parameters are the same as those used in Figure 2c.

**Figure 4 nanomaterials-12-03357-f004:**
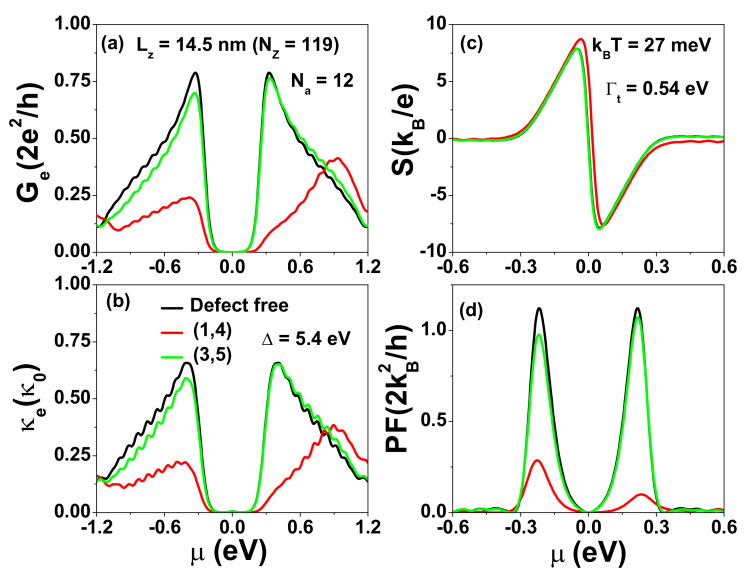
(**a**) Electrical conductance $G_{e}$, (**b**) electron thermal conductance $\kappa_{e}$, (**c**) Seebeck coefficient *S*, and (**d**) power factor $PF=S^{2}G_{e}$ as functions of μ for defect locations at $k_{B}T=27$ meV. The tunneling rates used are $\Gamma_{t}=0.54$ eV. The length of SGQD is $L_{z}=14.5$ nm ($N_{z}=119$). Each GQD in the SGQD structure has size $N_{a}=12$ and $N_{z}=7$.

**Figure 5 nanomaterials-12-03357-f005:**
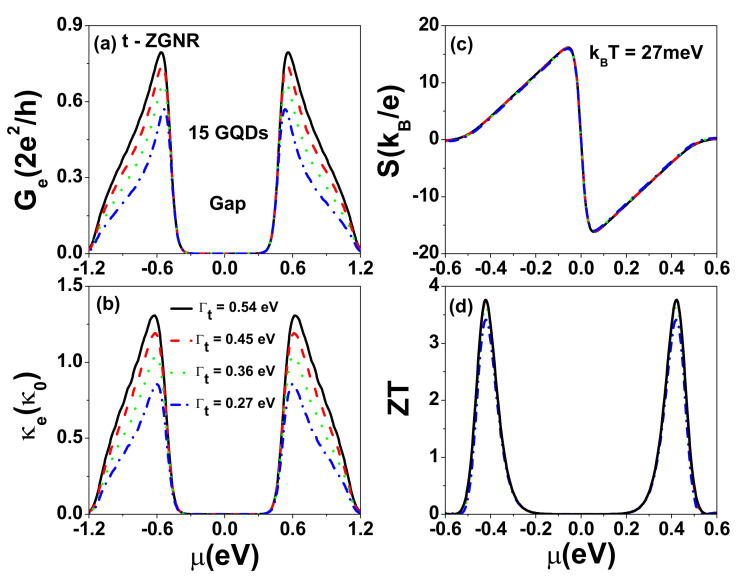
(**a**) Electrical conductance $G_{e}$, (**b**) electron thermal conductance $\kappa_{e}$, (**c**) Seebeck coefficient *S*, and (**d**) figure of merit ZT as functions of μ for various tunneling rates ($\Gamma_{t}$) at $k_{B}T=27$ meV. The t-ZGNR structure with $L_{z}=14.5$ nm considered is illustrated in Figure 1b. The size of each GQD in the t-ZGNR is characterized by $N_{a}=8$ and $N_{z}=7$.

**Figure 6 nanomaterials-12-03357-f006:**
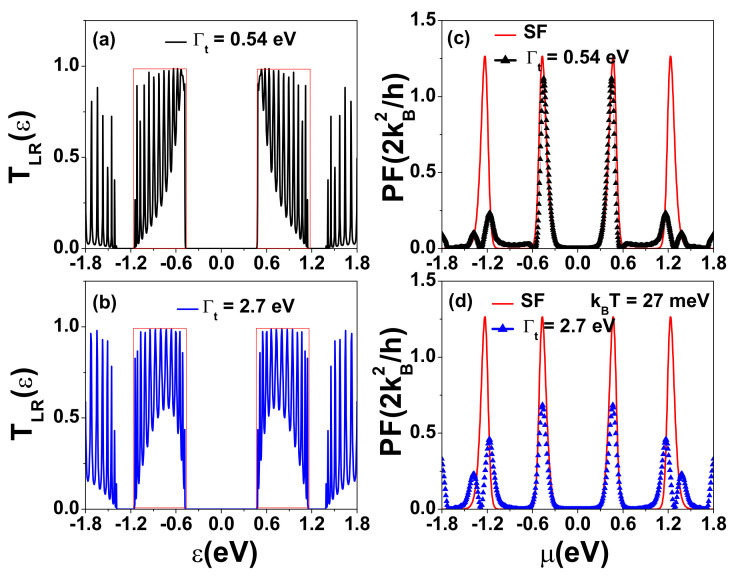
(**a**,**b**) Transmission coefficients $T_{LR}\left(\varepsilon \right)$ as functions of energy for a t-ZGNR with $\Gamma_{t}=0.54$ eV and $\Gamma_{t}=2.7$ eV. The $T_{LR}\left(\varepsilon \right)$ with square form (SF) is also plotted by a curve with red color. (**c**,**d**) Power factor as functions of μ for two tunneling rates at $k_{B}T=27$ meV. Other physical parameters are the same as those used in Figure 5.

**Figure 7 nanomaterials-12-03357-f007:**
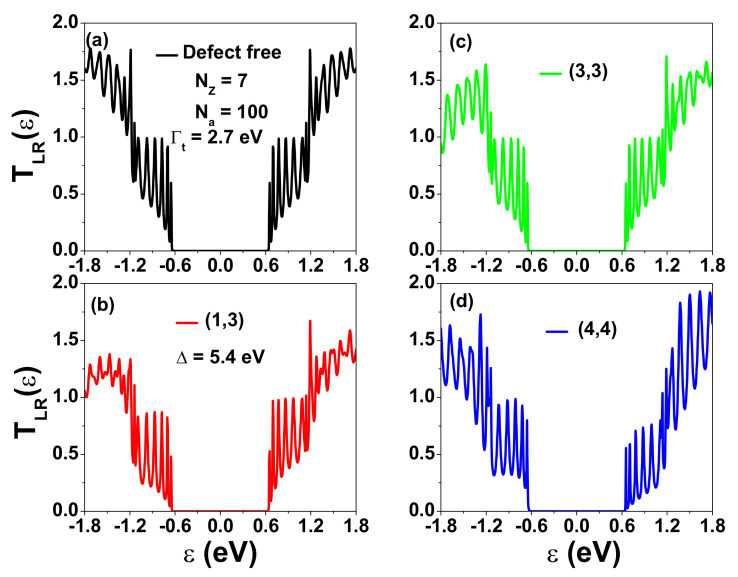
Transmission coefficient of AGNRs as a function of ε for $\Gamma_{t}=2.7$ eV, $N_{z}=7$, and $N_{a}=100$ ($L_{a}=10.5$ nm) with (**a**) no defect, (**b**) a defect at site (1,3), (**c**) a defect at site (3,3), and (**d**) a defect at site (4,4). The energy shift at a defect site adopted is Δ=5.4 eV.

**Figure 8 nanomaterials-12-03357-f008:**
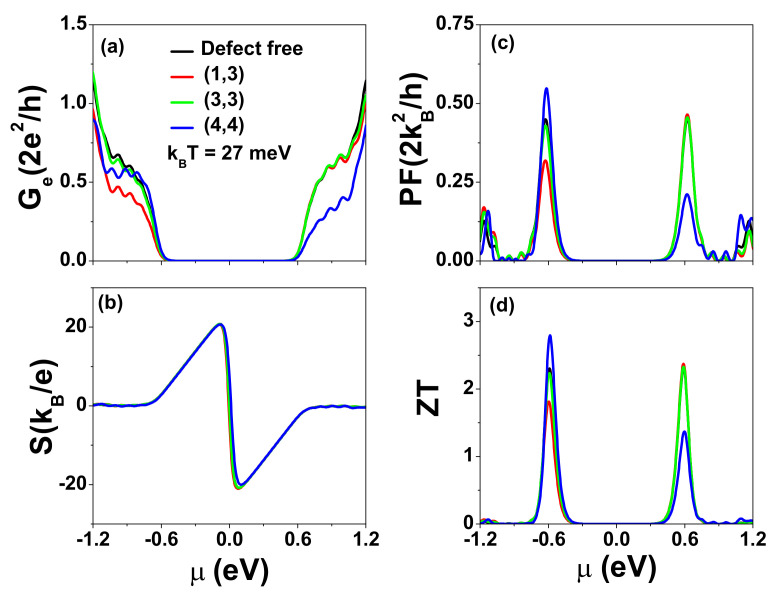
(**a**) Electrical conductance $G_{e}$, (**b**) Seebeck coefficient *S*, (**c**) power factor PF, and (**d**) figure of merit ZT as functions of μ for different defect locations at $k_{B}T=27$ meV. Other physical parameters are the same as those of Figure 7. The phonon thermal conductance ($\kappa_{ph}$) considered is the same as that used in Figure 5 ($\kappa_{ph}=0.0285$ nW/K).

**Figure 9 nanomaterials-12-03357-f009:**
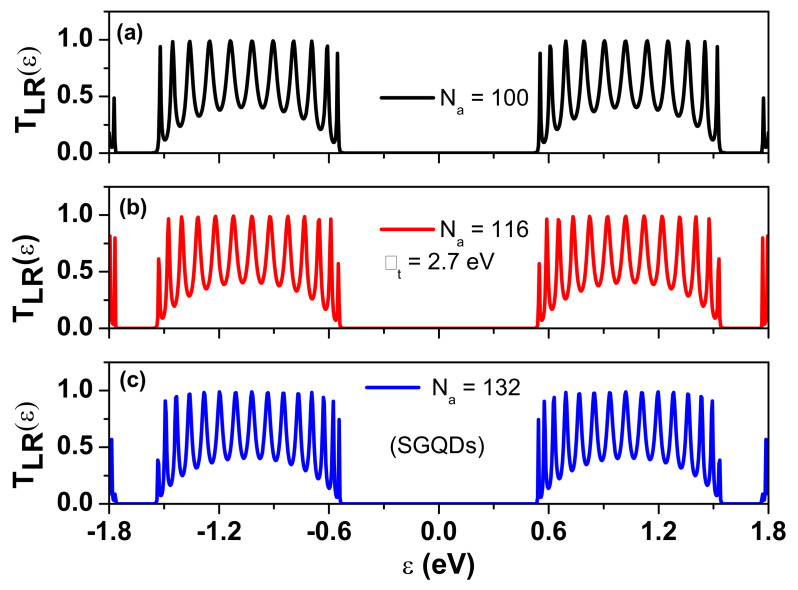
Transmission coefficient of SGQD with $N_{z}=5$ for various $N_{a}$ values as a function of energy ε. $\Gamma_{t}=2.7$ eV. Here, $N_{a}=12+8\times N_{QD}$, where $N_{QD}$ is the number of GQDs in the interior region of the structure, and they are sandwiched between two smaller GQDs of $N_{a}=6$ connected to electrodes. In (**a**–**c**) we consider cases with $N_{a}=100,116$, and 132 that correspond to $N_{QD}=11,13$, and 15, respectively. For $N_{a}=116$ and $N_{a}=132$, the lengths of SGQDs are 12.2 nm and 13.9 nm, respectively.

**Figure 10 nanomaterials-12-03357-f010:**
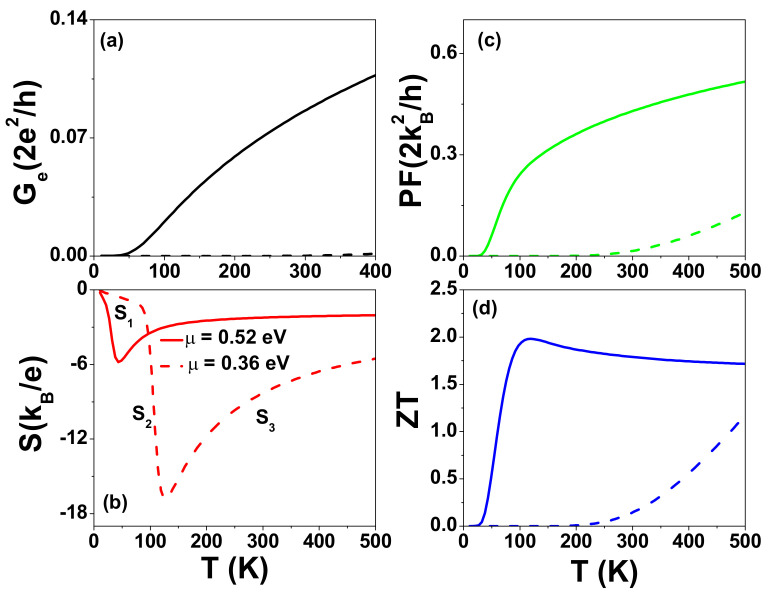
(**a**) Electrical conductance $G_{e}$, (**b**) Seebeck coefficient *S*, (**c**) power factor PF, and (**d**) figure of merit ZT of SGQD with $N_{z}=5$ and $N_{a}=132$ ($N_{QD}=15$) as functions of temperature for different μ at $\Gamma_{t}=2.7$ eV. The phonon thermal conductance ($\kappa_{ph}$) adopted is the same as in Figure 8.

## Data Availability

The data presented in this study are available upon reasonable request.

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
