# Peer review of "Contact Effects on Thermoelectric Properties of Textured Graphene Nanoribbons"

_nanomaterials, 2022, doi:10.3390/nano12193357_

Round 1
Reviewer 1 Report
The thermoelectric properties of graphene are studied intensively and many experimental results are available in the literature.The authors have made no effort to compare their theoretical findings with these experimental results. Moreover,at the present state-of -the art of graphene technology such nanostrucures presented in the manuscript are difficult if not impossible to be fabricated and tested.
Author Response
We thank the referee for the critical review, which helps to improve our manuscript. Following the referee’s suggestion, we have included 8 references in the revised manuscript as listed below. We have added discussions to compare our theoretical results with these experimental findings.
[28] Zuev, Y. M.; Chang, W.; Kim, P. Thermoelectric and magnetothermoelectric transport measurements of graphene, Phys. Rev. Lett. 2009, 102, 096807.
[29] Wei, P.; Bao, W. Z.; Pu, Y.; Lau, C. N.; Shi, J. Anomalous thermoelectric transport of Dirac particles in graphene, Phys. Rev. Lett. 2009, 102.166808.
[30] Xu, Y.; Li, Z.Y.; Duan,W. H., Thermal and thermoelectric properties of graphene, Small 2014, 10, 2182.
[31] Wang, Y. H.; Yang, J.; Wang, L. Y.; Du, K.; Yin, Q.; Yin, Q. J. Polypyrrole/graphene/polyaniline ternary nanocomposite with high thermoelectric power factor, ACS Appl. Mater. Interfaces 2017, 9, 23, 20124.
[32] Li, M.; Cortie, D. L.; Liu, J. X.; Yu, D. H.; Islam, SMKN; Zhao, L. L.; Mitchell, DRG; Mole, R. A.; Cortie, M. B.; Dou, S. X. Ultra-high thermoelectric performance in graphene incorporated Cu2Se: Role of mismatching phonon modes, Nano energy
2018, 53, 993.
[33] Ghosh, S.; Harish, S ; Ohtaki, M. ; Saha, B. B. Thermoelectric figure of merit enhancement in cement composites with graphene and transition metal oxides, Material today energy, 2020, 18, 100492.
[34] Wang Y.Y.; Chen D. R.; Wu J. K.; Wang T. H.; Chuang C. S.; Huang S. Y.; Hsieh W. P.; Hofmann M.; Chang Y. H.; Hsieh Y. P. Two-dimensional mechano-thermoelectric heterojunctions for self-powered Strain Sensors, Nano Lett. 2021, 21, 6990.
[70] Suarez, F.; Nozariasbmarz, A.; Vashaee, D.; Ozturk, M. C. Designing thermoelectric generators for self-powered wearable electronics, Energy Environ Sci, 2016, 9, 2099.
We cited these references in the introduction section (see revised manuscript)
“Several experimental studies of thermoelectric properties of graphene-related materials have been reported in recent years [28-34].”
and discussed the comparison with some of these references in Sec. 3.2.
“It is noted that the maximum S reaches 1.51mV/K [see Fig 5(c)], which is much
larger than that observed experimentally in gapless graphene [28,29].”
We also discussed relevant findings in these reference and how they can improve the thermoelectric performance of graphene heterostructures.
“The measured can be reduced from 300-600nm in a single layer graphene to 10nm in graphene nanostructures (see [31] and references therein).
Recently, very short have been reported experimentally, which offers promise for enhancing the figure of merit (ZT) of graphene heterostructures [30-34].”
We also cited Ref. [70] in the conclusion section. Ref. [70] discussed how to design thermoelectric power generators for applications in low-power electronics. Our finding is useful to provide electrical power of wearable electronics such as the new generation i-watch. In the conclusion of the revised manuscript, we wrote
“At room temperature, T=324K (), the electrical power output can reach 0.212nW/K for each SGQD implemented by using t-ZGNR. For an SGQD array with density of and ZT larger than 3, the power output is around mW/K, which can be applicable for low-power wearable electronic devices [70].”
Regarding the comment on “Moreover, at the present state of the art graphene technology such nanostructures presented in the manuscript are difficult if not impossible to fabricated and tested”, we fully appreciate the difficulty in fabricating high-quality t-ZGNR to realize the predicted effect. However, according to the reports in references [18-25] and the measurement techniques described therein, in particular the impressive achievement from Professor Crommie’s group, we believe that there is a good chance to implement the proposed t-ZGNRs and t-GANRs in the future.

Reviewer 2 Report
The ballistic transport and thermoelectric properties of finite textured graphene nanoribbons were theoretically investigated. The more favorable configuration has been proposed, i.e. tailoring zigzag nanoribbons in comparison with armchair nanoribbons
The coupling with the electrodes was performed using a bottom-up synthesize technique. Several findings are useful for the community. Minibands can be formed in these t-GNRs. Moreover, the energy gaps can be tuned by varying the size of the graphene quantum dots. Subband width and central gap of nanoribbons can be modulated by varying the size of quantum dots and inter-dot coupling strength
The paper is well organized and illustrated and the main conclusions are understandable even for a non-thetoretician. As a non-specialist I cannot give an opinion about the calculation method which involved a Green’s function approach within a tight-binding mode.
The results are compared to the previous theoretical studies highlighting the effect of defects when located either at the edge or in the interior region.
Author Response
We appreciate the positive comments from the referee. In the revised manuscript (see attachment), we have added 8 new references to include relevant experimental studies. References [28-34] presented experimental studies of thermoelectric properties of a single layer graphene and graphene junctions. Ref. [70] provides guidelines on the design of thermoelectric generators for self-powered wearable electronics. According to the suggestions in ref. [70] our finding may be useful for finding a suitable candidate for the application in low-power wearable electronics. We have also corrected some typos and improved the English.

Reviewer 3 Report
Paper devoted to investigation of contact effects on thermoelectric properties of textured graphene nanoribbons. Authors studied transport and thermoelectric properties of finite textured graphene nanoribbons (t-GNRs) connected to electrodes with various coupling strengths are theoretically studied in the framework of the tight-binding model and Green’s function approach. Due to quantum constriction induced by the indented edges, such t-GNRs behave like serially-coupled graphene quantum dots (SGQDs). Effects of defects and junction contact on electrical conductance, Seebeck coefficient, and electron thermal conductance of t-GNRs are calculated. When a defect occurs in the interior site of textured ZGNRs (t-ZGNRs), the maximum power factor within the central gap or near the band edges is found to be insensitive to the defect scattering. Furthermore, it was found that SGQDs formed by t-ZGNRs have significantly better electrical power outputs than those of textured ANGRs due to the improved functional shape of the transmission coefficient in t-ZGNRs.
The article is interesting and relevant, but there are the following comments:
- The purpose of the work should be formulated separately in the introduction.
- In conclusion, it is necessary to indicate the possibility of practical use of the results.
After these corrections, the paper can be published in Nanomaterials.
Paper devoted to investigation of contact effects on thermoelectric properties of textured graphene nanoribbons. Authors studied transport and thermoelectric properties of finite textured graphene nanoribbons (t-GNRs) connected to electrodes with various coupling strengths are theoretically studied in the framework of the tight-binding model and Green’s function approach. Due to quantum constriction induced by the indented edges, such t-GNRs behave like serially-coupled graphene quantum dots (SGQDs). Effects of defects and junction contact on electrical conductance, Seebeck coefficient, and electron thermal conductance of t-GNRs are calculated. When a defect occurs in the interior site of textured ZGNRs (t-ZGNRs), the maximum power factor within the central gap or near the band edges is found to be insensitive to the defect scattering. Furthermore, it was found that SGQDs formed by t-ZGNRs have significantly better electrical power outputs than those of textured ANGRs due to the improved functional shape of the transmission coefficient in t-ZGNRs.
The article is interesting and relevant, but there are the following comments:
- The purpose of the work should be formulated separately in the introduction.
- In conclusion, it is necessary to indicate the possibility of practical use of the results.
Author Response
We thank the referee for these helpful comments. We have followed the referee’s suggestions and revised the manuscript accordingly.
Answer to Point 1: In the revised version, we have revised the last paragraph of introduction section as follows.
“Several experimental studies of thermoelectric properties of graphene-related materials have been reported in recent years [28-34]. Nevertheless, there is a paucity of studies to consider the contact effect on thermoelectric properties of t-AGNRs and
t-ZGNRs [35]. In this work, our goal is to optimize the transmission coefficient of t-GNRs with the length short than , but larger than under different coupling strengths with the electrical contact."
Answer to Point 2: In the conclusion of the revised manuscript, we added: “At room temperature, T=324K (), the electrical power output can reach 0.212nW/K for each SGQD implemented by using t-ZGNR. For an SGQD array with density of and ZT larger than 3, the power output is around mW/K, which can be applicable for low-power wearable electronic devices [70].”
In this revised manuscript, we added 8 new references to include relevant experimental studies. References [28-34] presented experimental studies of thermoelectric properties of a single layer graphene and graphene junctions. Ref. [70] provides guidelines on the design of thermoelectric generators for self-powered wearable electronics. According to the suggestions in ref. [70] our finding may be useful for finding a suitable candidate for the application in low-power wearable electronics. We have also corrected some typos and improved the English.

Reviewer 4 Report
The article is very nice and it is worth to be published.
Author Response
We thank the referee for the positive comments. We have further improved the manuscript by adding eight references in the updated version. These references consist of experimental measurements of thermoelectric properties of graphene [28-34,70] and discussed the possible applications of graphene-based thermoelectric devices [70] which are relevant to our current study. We have also corrected some typos and improved the English.

Round 2
Reviewer 1 Report
The manuscript was improved. The lack of the experiemntal evidence is the main problem unsolved.